# Diagnostic Value of Frequently Implemented Provocative Tests in the Assessment of Shoulder Pain—A Glimpse of Current Practice

**DOI:** 10.3390/medicina57030221

**Published:** 2021-03-01

**Authors:** Nahum Rosenberg

**Affiliations:** Laboratory of Musculoskeletal Research, Rambam—Health Care Campus and Ruth and Bruce Rappaport Faculty of Medicine, Technion—Israel Institute of Technology, 8 HaAliya HaShniya St., POB 9602, Haifa 3109601, Israel; nahumrosenberg@hotmail.com; Tel.: +972-54-4685130; Fax: +972-4-8590930

**Keywords:** diagnostic odds ratio, likelihood ratio, provocative test, shoulder joint, shoulder pain

## Abstract

Initial evaluation of chronic shoulder disability is a diagnostic challenge due to the anatomic complexity of the shoulder joints. For this purpose, several diagnostic tools utilizing provocative testing exist, but only a few have a reliable basis for their diagnostic value. Therefore, objectively determining the predictive value of these tests in identifying the precise anatomical source for disability—subacromial, intraarticular or other—is essential in order to proceed with further imaging evaluations for final objective diagnoses. Using validated clusters of provocative tests should improve their diagnostic values.

Shoulder pain and restriction usually originate from intraarticular and/or subacromial irritation and can also be referred to cervical or visceral sources. The pathological process that causes shoulder pain and restriction is usually due to inherent anatomical abnormalities or those acquired after injury and/or surgery or following intrinsic degenerative or inflammatory tendinopathy or arthropathy. Therefore, it is logical that a patient suffering from shoulder pain and restriction, independently from the previously known shoulder pathology, should be evaluated initially by two main diagnostic tools. Firstly, a general evaluation should be undertaken of the pain source localization in the shoulder, which can be done using diagnostic scores and provocative tests; this might make it possible to localize the pain and restriction, often to either the intraarticular glenohumeral or subacromial anatomical regions. Subsequentially, a final diagnosis should be verified using more objective imaging modalities. Imaging provides the diagnosis of the intraarticular and/or extraarticular pathology in the shoulder. Naturally, neither of the current modalities can precisely indicate the sole source of pain; therefore, further modalities, such as directed injections of anesthetic, might help in obtaining more precise diagnoses.

However, due to the shoulder’s anatomic complexity, an initial diagnosis by physical examination seldom has a clear prognostic value. Accordingly, numerous provocative tests have been proposed and used but most lack a clear and proven diagnostic value since reported studies on their diagnostic values are mostly statistically underpowered.

The verification of the source of shoulder pain is an important diagnostic challenge. Shoulder pain prevalence, with or without previous history of trauma, is high, being reported in 50% of the general population [1], and is accompanied by a high degree of disability. Aiming at facilitating the initial diagnosis, numerous provocative diagnostic tests have been suggested. Provocative tests are designed to address the anatomical structure suspected of being the source of the pain and the shoulder complex’s restriction. Since provocative tests for painful shoulder evaluation are easy to perform, they should play an important role in generating an initial “working diagnosis” of shoulder pain or restriction, in order to determine the appropriate imaging modality required to provide the final diagnosis; the treatment plan should be established accordingly. This approach is clinically logical since shoulder disability can originate from numerous intraarticular and extraarticular structures. Therefore, a uniform imaging approach is inapplicable for the initial shoulder disability evaluation, which should rather be tailored individually according to the prior “working” diagnosis.

When relying on one or more provocative tests, their diagnostic value should be verified. This diagnostic value is usually based on validated clinical studies that determine the specificity and sensitivity of a specific test or scoring method. These values can further determine the positive and negative likelihood ratios and the diagnostic odds ratio (LR+, LR−, and DOR, respectively) that indicate whether the implemented clinical test can be a reliable basis for a substantial “working diagnosis” [2].

Although numerous theoretically sound provocative tests have been suggested for the evaluation of specific pathological sources of shoulder pain and restriction, most were found to be of a limited or even marginal diagnostic value, i.e., LR+ below a value of 5 or even below a value of 2 and LR− above a value of 0.1 or even above a value of 0.2 [3]. These insufficient characteristics of the tests used have an undesired impact on the decision-making process in the initial evaluation of shoulder disability. Several solutions have been proposed to overcome this diagnostic difficulty, such as the clustering of different provocative tests and scores related to a specific diagnosis [4], standardized isometric strength measurements of rotator cuff muscles aimed at distinguishing between their intrinsic and extrinsic pathology [5], and validated scoring systems of shoulder functions for a gross identification of shoulder instability vs. degenerative shoulder pathology [6,7].

In chronic cases, the most important diagnostic uncertainty of the initial shoulder evaluation concerns the exact source of the shoulder disability. Even when the initial acute diagnosis or chronic pathology is known—such as fracture, shoulder dislocation, or primary or secondary intraarticular degeneration—there is always the basic diagnostic uncertainty as to what the actual source of the disability is, i.e., intraarticular, subacromial, both, or another source. Currently, for most shoulder pathologies, it is not clear which provocative tests have sufficient diagnostic value for meaningful decision-making. There are not enough studies assessing statistically adequate cohorts of patients with chronic glenohumeral intraarticular or extraarticular pathologies to provide meaningful information about the predictive diagnostic values of commonly used provocative tests designed for glenohumeral and subacromial evaluation. The commonly used provocative tests for intraarticular evaluation are Speed’s test for the tearing or irritation of the long head of the biceps tendon; Duga’s test for anterior glenohumeral dislocation; O’Brien’s active compression test for superior labral tears; the anterior apprehension test for labral tears; and Gerber and Ganz’s anterior drawer test for anterior shoulder instability due to an insufficiency of the anterior capsule complex. The common provocative tests for subacromial evaluation include Codman’s sign (the drop arm sign) for supraspinatus tendon tears; Jobe’s supraspinatus test for *supraspinatus* tendon tears or irritation; the anterior apprehension test for subacromial irritation; Dawbarn’s sign for subacromial bursitis; Gerber’s lift-off test for *subscapularis* tendon tears or irritation; the supine Napoleon test (belly-press test) for *subscapularis* tendon tears; the painful arc test for subacromial impingement; and Neer’s impingement test for subacromial impingement. We cannot expect to get a precise diagnosis of a specific pathology using provocative tests. Still, we can define a general direction for the initial diagnosis, i.e., intraarticular or subacromial, that should lead to appropriate further imaging. Therefore, it is imperative to know which clinical evaluations from among the above-mentioned provocative tests are effective for further verification using the more objective imaging modalities (X-rays, ultrasound scans, MRI scans, etc.) and thus for the final diagnosis determination of the source of shoulder disability. Thus, reliable LR+, LR−, and DOR values for each test and scoring system should be known for the efficient use of these diagnostic modalities [8]. The diagnostic relevance of LR values is graded as conclusive (LR+ > 10.00; LR− < 0.10), moderate (LR+ in the range of 5.00–10.00; LR− in the range of 0.10–0.20), marginal (LR+ in the range of 2.00–5.00; LR− in the range of 0.2–0.50), and non-diagnostic (LR+ in the range of 1.00–2.00; LR− in the range of 0.50–1.00) [3]. The DOR values, which indicate the discriminatory diagnostic power of the tests, can be defined as non-diagnostic (equal to and below the value of 1.00), low diagnostic (the range of values is above 1.00 and below 3.00), moderately diagnostic (the range of values is 3.00–30.00), and highly discriminatory for diagnosis (values are above 30.00). ROC curves for the pooled tests and scores for each study group can provide the diagnostic value of a cluster of tests and scores by determining the breaking point that has the highest derivative value on the curve that identifies the optimal additive diagnostic value of each cluster (4).

Most reports on the diagnostic value of provocative tests in evaluating shoulder disability are based on small cohorts of patients (below 80 individuals) [4,9]. This might be why the inconclusive LR+, LR−, and DOR values are present: due to the low power of the diagnostic accuracy of the sensitivity and specificity calculations [10]. Since the general prevalence of shoulder pain and restriction is around 50%, a study of a group of patients for the investigation of an even marginal diagnostic meaning of the diagnostic tests should exceed 160 patients for a minimal LR+ of 2, and should include at least 228 patients for a maximal LR− of 0.4 [11]. Therefore, the marginally meaningful results for evaluating these tests’ diagnostic relevance should involve more than 230 patients.

I am aware of only three published reports that could be included in a systematic review of the accuracy of provocative tests of the shoulder and have sufficient power for meaningful clinical conclusions according to the abovementioned criteria [9]: a study that investigated a cohort of 256 patients who were evaluated for suspected SLAP lesion and presented substantial evidence for the high diagnostic value of the active compression test (LR+ = 61, LR− = 0) [12]; a study of 303 patients that showed the prognostic insufficiency of the Crank test (anterior apprehension test) for SLAP diagnosis (LR+ = 1, LR− = 1) [13]; and an additional study on the same subject that investigated 544 patients and showed insufficient diagnosis values for Neer’s test (LR+ = 1, LR− = 1), the Hawkins test (LR+ = 1.2, LR− = 0.9), and the painful arc test (LR+ = 1.2, LR− = 0.8) [14].

Therefore, according to currently available published data, a specific further imaging evaluation cannot adequately rely on the initial assessment provided by most of the provocative tests mentioned above. In other words, clusters of tests can be useful for giving a general indication of the source of pain and disability localized in the shoulder but cannot point out the exact anatomical source of the pathology. This fact should be considered in clinical decision-making regarding the evaluation and diagnosis of the painful and restricted shoulder. Most likely, commonly used provocative tests are not diagnostically sufficient for this purpose. Therefore, it is imperative to gain more data from sufficiently statistically powered studies and determine a concise list of proven practical provocative tests with reliable diagnostic values for painful shoulder evaluation.

## Data Availability

Not applicable.

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
