# Peer review of "Diagnostic Value of Frequently Implemented Provocative Tests in the Assessment of Shoulder Pain—A Glimpse of Current Practice"

_medicina, 2021, doi:10.3390/medicina57030221_

Round 1

Reviewer 1 Report

Thank you very much for giving me the oppourtunity to review this article presenting the opinion of the author according the diagnostic values of frequently implemented provocative tests in the assessment of shoulder pain. First of all, the article is clearly presented and adresses a relevant topic. However, being only a "comment" the scientific value is limited, since it only encourages to the reader to conduct further studies in order to gain validated data and present the dilemma of shoulder test. Furthermore, I assume a lot of shoulder surgeons are aware of limited diagnostic values of the provocative tests. However, it might be interesting for the readership of general medicine.

There are only a few points which need to be adressed.

a)I agree with the author, that shoulder tests need to be interpreted in a cluster and maybe you should emphasize this point a little bit more. 

b) "For example, the commonly used provocative tests for intraarticular evaluation are: Speed’s test for the long head of the biceps tendon tear or irritation; Duga’s test for anterior glenohumeral dislocation; O’Brien’s active compression test for superior labral tears; the anterior apprehension test for labral tear; and Gerber-Ganz’s anterior drawer test for anterior shoulder instability due to insufficiency of the anterior capsule complex. The common provocative tests for subacromial evaluation include: Codman sign (drop arm sign) for supraspinatus tendon tear; Jobe’s supraspinatus test for supraspinatus tendon tear or irritation; the anterior ap-prehension test for subacromial irritation; Dawbarn’s sign for subacromial bursitis; Ger-ber’s lift-off test for subscapularis tendon tear or irritation; Supine Napoleon’s test (belly-press test) for subscapularis tendon tear; the painful arc test for subacromial impingement; and Neer’s impingement test for subacromial impingement. "

--> I would recommend to shorten this abstract. Maybe give only one example.

c)

"Therefore, it is imperative to know what clin-ical evaluation by the above-mentioned provocative tests is effective for further verifica-tion using the more objective imaging modalities (X-rays, ultrasound scans, MRI scans, etc.) for the final diagnosis determination of the source of shoulder disability. "

--> PLease emphasize this point further.

Reviewer 2 Report

In this paper, authors present an opinion about  Diagnostic value of frequently implemented provocative tests in the assessment of shoulder pain – a glimpse of current practice.
  Provocative tests are designed to address the anatomical structure suspected of being the source of pain and restriction of the shoulder complex. Since provocative tests for painful shoulder evaluation are easy to perform, they should be important for generating an initial “working diagnosis” of shoulder pain or restriction to determine the appropriate imaging modality required further that would provide the final diagnosis, and the treatment plan could be established accordingly. 

Reviewer 3 Report

The author submitted a nice, narrative review/opinion focusing on diagnostic provocative tests used in the diagnostics of shoulder pain. Overall this is a nice paper written by an experienced clinician. I really did enjoy reading it and agree with the vast majority of the authors statements, however as every narrative paper it is biased to some degree. Consequently I have two major and one minor suggestions for the author :

First major point - I strongly agree with the author that the classical history taking / physical examination / imaging studies algorithm has major limitations, specifically in case the of shoulder. Many provocative tests used in shoulder surgery have limited accuracy, and imaging studies are now indispensable in the diagnostic protocol. However as technology develops, certain imaging modalities – especially Ultrasound became more and more important in everyday practice of orthopaedic surgeons (not just radiologists) and a growing number of shoulder surgeons incorporate ultrasound imaging to their routine examination protocol. I think that in the hands of an experienced surgeon this technique combines the best of both worlds. I actually se an analogy to the situation which was observed in paediatric orthopaedics when ultrasound imaging was introduced to DDH diagnostics in 1980s/1990s. Unreliable clinical tests supported by imaging studies revolutionized this field, and I believe that perhaps a less spectacular but significant improvement can be observed in shoulder surgery – please consider discussing this aspect in the paper, perhaps at the end of the paper ? Perhaps we need studies combining these tests with sonographic data and MRI or arthroscopic data ? Perhaps we need good standardized protocols like they use in DDH diagnostics, or echocardiography ?

The other major point – I get the impression that you consider imaging studies as the “golden standard” and suggest that they ultimately confirm the diagnosis, you even describe them as objective at some point. I would suggest toning this down a little bit. There is evidence that various pathologies such tendon disorders may appear simultaneously in different localization of the shoulder. Rotator cuff tendinopathy and tears are the most common ones among them and they are usually associated with the long head of the biceps tendon (LHBT) pathology, superior labrum anterior to posterior (SLAP) injuries, subacromial impingement syndrome and acromioclavicular joint (ACJ) disorders. Complex and multi-tendon shoulder injuries significantly complicate the process of diagnosis, treatment and rehabilitation and there is often a difficulty in pinpointing one specific source of pain. Additionally in many cases there is severe discrepancy between clinical, imaging and histological findings and various demographic parameters may also play a role. Please consider including discussing this in 2-3 sentences and including following open-access papers : https://doi.org/10.3390/jcm10040599, . doi: 10.5603/FM.a2017.0093

The last small point : the paragraph which includes positive and negative likehood ratios … (LR+,LR-, DOR) – please consider rewriting it – with full names for clarity : positive likelihood ratio (LR+), negative likelihood ratio (LR-), diagnostic …  then my suggestion would be insert the paragraph where the meaning of LR/DOR values is described as meaningful/moderate/mariginal etc (in the submitted version these parts are separated by the description of various clinical tests)
